# Perfect Combination of LBL with Sol–Gel Film to Enhance the Anticorrosion Performance on Al Alloy under Simulated and Accelerated Corrosive Environment

**DOI:** 10.3390/ma13010111

**Published:** 2019-12-25

**Authors:** Xia Zhao, Shuai Yuan, Zuquan Jin, Binbin Zhang, Nazhen Liu, Shibo Chen, Shuan Liu, Xiaolin Sun, Jizhou Duan

**Affiliations:** 1Key Laboratory of Marine Environmental Corrosion and Bio-fouling, Institute of Oceanology, Chinese Academy of Sciences, Qingdao 266071, China; ys520399@163.com (S.Y.); zhangbinbin@qdio.ac.cn (B.Z.); nazhenliu@163.com (N.L.); liushuan@nimte.ac.cn (S.L.); duanjz@qdio.ac.cn (J.D.); 2Cooperative Innovation Center of Engineering Construction and Safety in Shandong Blue Economic Zone, Qingdao University of Technology, Qingdao 266032, China; sunxiaoxiao0526@163.com; 3Open Studio for Marine Corrosion and Protection, Pilot National Laboratory for Marine Science and Technology, Qingdao 266235, China; 4Center for Ocean Mega-Science, Chinese Academy of Sciences, Qingdao 266071, China; 5School of Chemistry and Chemical Engineering, Guangxi University, Nanning 530004, China; dfcsb@126.com

**Keywords:** multilayer, corrosion, impedance, layer-by-layer

## Abstract

Given their outstanding versatile properties, multilayered anticorrosion coatings have drawn great interest from researchers in the academic and engineering fields. However, the application of multilayered coatings is restricted by some limitations such as low interlayer compatibilities, the harsh preparation process, etc. This work introduced a composite film fabricated on a 2A12 aluminum alloy surface, including an anodic oxide film, a sol–gel film, and a layer-by-layer (LBL) self-assembling film from bottom to top. The microstructure and elemental characterization indicated that the finish of the coating with the LBL film resulted in a closely connected multilayered coating with a smoother surface. The anticorrosion performance was systematically evaluated in the simulated corrosive medium and neutral salt spray environment. The integrated coating with the LBL film presented an excellent anticorrosion ability with system impedance over 10^8^ Ω·cm^2^ and a self-corrosion current density two orders of magnitude lower than that of the other coatings. After the acceleration test in a salt spray environment, the multilayered coatings could still show a good protective performance with almost no cracks and no penetration of chloride ions. It is believed that the as-constructed multilayered coating with high corrosive properties and a fine surface state will have promising applications in the field of anticorrosion engineering.

## 1. Introduction

Aluminum and its alloys are widely used in the fields of shipbuilding, oceanography engineering, aerospace, and machine manufacturing because of its low density, low thermal expansion coefficient, high strength, and strain performance [1,2], which have drawn much attention from both researchers and engineers as promising materials for electrical appliances and the machinery industry [3,4,5,6,7,8]. Under certain circumstances, Al alloys possess some anticorrosion resistance because of the naturally formed oxidized thinner film [9,10]. However, the natural thin and inhomogeneous oxide film dissolves substantially when exposed to corrosive environments containing aggressive chloride ions, resulting in localized corrosion, mechanical failure, considerable financial cost, and even catastrophic accidents [11,12].

In order to improve the corrosion resistance and stability of the Al alloys, more protective technologies need to be explored to extend their service life in some special environment, such as marine, industry, etc. [13,14]. There are two typical approaches to improve the corrosion resistance of Al alloy; one is to ameliorate the element proportion of Al alloys, and the other is to cover a protective film on the surface of Al alloys. At present, artificial fabricated films upon Al alloy substrates are commonly used to improve the protective ability, including anodic oxidation [15,16], laser cladding [17], the sol–gel method [18], self-assembly technology [19,20], anticorrosion coatings [21] etc.

The sol–gel method is regarded as an effective and environmentally friendly coating sealing technology. This method has the advantages of low costs, simple operation, and low reaction temperature. Dense and porous film can be obtained from this technology, which can prolong the service life and enlarge the application of Al alloys in the marine environment [22,23,24,25,26,27,28]. Salazar-Hernández et al. presented the DBTL behavior in alkylalkoxisilane condensation as anticorrosive coatings in Al-6061 for aerospace applications. This study was researched on crack propagation control by using DBTL as polycondensation catalyst, and it could be possible to extend the use of this catalyst within the sol–gel process [29]. However, due to improper reaction conditions in the heat treatment process, the film could crack and initiate the consequent degradation of the protective film [30]. Therefore, it is extremely urgent to enhance the anti-cracking performance of the sol–gel film, so as to improve its anticorrosive ability. 

Recently, a layer-by-layer (LBL) self-assembly method has been developed as a promising technique for the synthesis of composite functional film with evident advantages, such as simplicity, versatility, controllability etc., attracting a great amount of interest from researchers as a strategy for creating anticorrosion films [31]. The functional film is composed of polyelectrolyte multilayers, which are formed by constructing the polyelectrolyte with positive and negative charges through electrostatic adsorption layer by layer onto the metal surface [32,33,34]. The LBL self-assembly technique is based on the consecutive adsorption of polyanions and polycations via electrostatic interactions [23,35,36]. At present, the widely used materials for LBL self-assembly are polyethylenimine (PEI) as polyanions and poly (acrylic acid) as polycations, which often exhibit exponential growth [37]. By applying the LBL self-assembly technique, polyelectrolyte molecules could adsorb on surface of the sol–gel film, repairing the defects such as cracks and microholes by electrostatic adsorption, obviously improving the compactness of the film. Therefore, the combination of these two methods could be used for the improvement of anticorrosion performance of Al alloy. At present, limited research about the combined technology can be found, compared to the single protective film with some microcracks, it would reduce the potential risk of corrosion in the future service.

In this paper, anticorrosive coatings were fabricated on Al alloy substrate, combining and utilizing the advantages of LBL and the sol–gel method. A porous oxide film, pore-sealing sol–gel film, and LBL self-assembly film were successively prepared. The surface morphologies, composition, and surface roughness of the films were comprehensively characterized, and the protective mechanism of the films was speculated. Equivalent circuits were established to simulate the anticorrosion performance of different films in NaCl solutions, and the electrochemical parameters were obtained. The morphologies and composition were compared after the salt spray test of the protective films to evaluate the long-term performance.

## 2. Experimental and Characterization

### 2.1. Materials and Reagents

The major elemental composition of Al alloys 2A12 are (wt %) Si 0.5, Cu 3.8–4.9, Mg 1.2–1.8, Zn 0.3, Mn 0.3–0.9, Ti 0.15, and Al balance. The material has a thickness of 3–4 mm and was purchased from Beijing Goodwill Metal Technology Co., Ltd., (Beijing, China). Acetone, ethanol, nitric acid, aluminum isopropoxide, propanol, and acetyacetone were all obtained from Sinopharm Chemical Reagent Co., Ltd., (Shanghai, China). Polyethyleneimine) (PEI) with a MW of 70 kDa and graphene oxide of 1 wt % were purchased from Shanghai Aladdin Bio-Chem Technology Co., Ltd., (Shanghai, China). Poly (acrylic acid) (PAA) with a MW of 450 kDa was purchased from Sigma-Aldrich (Shanghai, China). All the reagents were of analytical grade and used without further purification. Deionized water with a resistivity of 18.0 uS/cm was used in all the experiments.

### 2.2. Pretreatment and Preparation of Anodic Oxide Film

The Al alloy samples were hand-cut and punched. They were polished by a grinding wheel and premilling machine. A natural oxide film was formed by alkali washing in the air. Then, the samples were ultrasonic cleaned with absolute ethyl alcohol and washed with distilled water. Finally, they were polished with diamond polishing agent and dried with an air blower.

In the anodization experiment, the Al alloy sample was connected to the positive pole, and the platinum electrode was connected to the negative pole of a power source. The electrolyte for the anodization experiment contains 40 g/L of oxalic acid, the anodizing voltage was 55 V, the oxidation current density was less than 2 A/dm^2^, and the anodization time was 60 min. An ice water bath was used to guarantee the constant temperature of the electrolyte. After being washed with deionized water and dried with cold air, the anodized film on Al alloy was obtained. The samples were put into a desiccator with silica gel to avoid moisture.

### 2.3. Preparation of the Sol–Gel Film

Alumina sol was firstly prepared as follows: 2 g of aluminum isopropoxide was added to 50 mL of n-propanol under continuously stirring until the aluminum isopropoxide was completely dissolved. Then, 2 mL of acetylacetone and 200 mL of distilled water were synchronously added under stirring. Finally, 3–5 drops of nitric acid as stabilizer were added to the above solution, and then stirred for 2 h on a magnetic stirrer and aged for 24 h to obtain a clear and transparent alumina sol.

The sol–gel film was further prepared on the surface of the Al alloy with anodic oxide film. The anodized Al alloy sample was suspended perpendicularly to the surface of the alumina sol and immersed in the sol at a uniform rate. Then, the sample was allowed to stay for 1 min in the sol to ensure that the prepared sol–gel film was uniform. Then, the sample was pulled out from the alumina sol at a rate of about 20 mm/min, and a layer of sol–gel film was successfully covered on the Al alloy surface. The sol was quickly converted into a wet gel in the atmosphere. The sample was heated under 120 °C for 30 min and then cooled at the room temperature. This operation was repeated for six times to improve the corrosion resistance of the sample.

During the sol–gel preparation and dip-coating procedure, the room temperature and air relative humidity were maintained at (23 ± 2) °C and (55% ± 5%), individually.

### 2.4. Preparation of the LBL Self-Assembled Protective Film

The Al alloy sample sealed with sol–gel film was immersed in a beaker containing 4 mg/L PAA for 5 min, and dipped in distilled water for 2 min. The sample was subsequently placed into a beaker containing 4 mg/L PEI for 5 min and also dipped in distilled water for 2 min. This procedure was called one cycle, which was repeated for 10 times in total. Then, the polyethyleneimine and polyacrylic acid LBL self-assembled protective film, marked (PAA/PEI)_10_, was obtained. The whole procedure of film preparation was illustrated in Figure 1.

### 2.5. Methods

#### 2.5.1. Characterization

Scanning electron microscopy (SEM) (Hitachi S-3400N, Tokyo, Japan) and atomic force microscope (AFM) (Multimode 8, Bruker, Karlsruhe, Cermany) were employed to observe the surface morphology of the samples. The samples for SEM determination were sputtered with gold. The elemental analysis was characterized by energy dispersive X-ray spectrum (EDS). Atomic force microscopy (AFM) measurements were performed at room temperature in the air and operated in tapping mode. The tip model is SCANASYST-AIR and the material is silicon tip on a nitride lever. The scanning rate is 0.400 Hz, and the scan angle is 0°. The scanning area of AFM measurement was 10 μm × 10 μm and 1 μm × 1 μm. The chemical compositions were measured by XPS (ESCALAB250Xi, Thermo Scientific, Waltham, MA, USA) using a 58° take-off angle. The spectra were recorded with monochromatized Al kα radiation (1486.6 eV) as the excitation source with a base pressure of 1.3 × 10^−9^ mbar, at a constant power of 150 W (15 kV, 10 mA). The pass energy was 20 eV for the high-resolution spectra. The binding energy of adventitious carbon (C1s: 284.8 eV) was used as a basic reference.

#### 2.5.2. Electrochemical Measurements

Electrochemical impedance spectroscopy (EIS) measurements were performed using a P 4000+ electrochemical workstation (Princeton Applied Research, Princeton, NJ, USA). A classical three-electrode cell system was employed, in which a saturated calomel electrode (SCE) was used as the reference electrode, a platinum plate was used as the counter electrode, and the Al alloy samples were used as the working electrode, respectively. EIS measurements were performed over the frequency range from 100 kHz to 10 mHz. The EIS data was analyzed and fitted by electrical equivalent circuit models using the ZSimpWin software (V3-10, Echem Software, Ann Arobr, MI, USA).

#### 2.5.3. Salt-Spray Test

The neutral salt-spray test was conducted to evaluate the corrosion resistance of different samples covered with anodic oxide film, sol–gel film, and LBL self-assembling film according to GB/T 10125 standard in a salt-spray chamber (SUGA CYP-90, Tokyo, Japan). Then, 5 wt % NaCl solution with pH 6.5–7.2 was used as media, and the test temperature was (35 ± 2) °C. The spray pressure was between 0.1 and 0.15 MPa, and the inlet pressure was between 0.2 and 0.4 MPa. The panels with the size of 20 × 20 × 3 mm^3^ were placed in the chamber at an angle of 45°. Three duplicate specimens were tested for different film samples. The interval between each specimen was above 5 mm to avoid interaction. The morphology of the panels was recorded by a digital camera every 24 h, and the total experiment period was 168 h.

## 3. Results and Discussion

### 3.1. Surface Morphologies and Composition of the Films

Figure 2 showed the typical SEM images of a blank Al alloy and the different resultant films. On the surface of the untreated Al alloy (Figure 2a), numerous scratches and irregular porous structures were observed. The heterogeneous structure with special dislocations stress can induce a higher increment of pitting corrosion [38]; so, further treatment was needed.

The uneven surface of the Al alloy substrate was improved after the anodization process (Figure 2b). The enlargement image showed that the surface was covered with a porous film accompanied by some waves, which could be due to the simultaneous growth and dissolution during the anodization process. Although the anodic oxide film could provide certain protection for the substrate, the corrosive ions would stay in the microporous structure and cause local corrosion as exposed to the corrosive medium. Thus, some sealing methods were needed for improving the anticorrosive performance of the film. Figure 2c showed the surface morphology of a sol–gel film, which was prepared on the surface of the anodic oxide film. Except for some non-uniform bulk materials, a homogeneous surface with less micropores was observed on the sol–gel film. The better surface state indicated that the set temperature of the drying heat treatment was appropriate during the preparation process of the sol–gel film. The aluminum sol could penetrate into the micropores and cracks of the anodic oxide film by physical adsorption, and then filled and sealed the defects. The sealing effect was obvious, and the anticorrosive ability of Al alloy was expected to be improved. However, it can be seen that the sol–gel film was not ideal. Some slight cracks still existed on the surface, which could be due to the volatilization of organic molecules dissolved in the sol–gel solution in the process of drying and heat treatment and film shrinkage.

Figure 2d showed the surface morphology of the LBL self-assembly film. It was covered with a relatively uniform and dense material, including some waved structure, and defects, such as cracks and pores, was not observed. The surface of the film was smoother and more uniform than the other films. The polyanion-electrolyte ions was successfully adsorbed and covered on the surface of the sol–gel film by self-assembly technology. The compactness of the film was improved, which could protect the Al alloy from the corrosive medium directly, and then enhance its anticorrosion performance. According to the EDS results, the content of the C element in the LBL film was increased by four times compared with that of the sol–gel film. The N element, which was not included in other films, appeared in the LBL film with the content of 9.08%, indicating the existence of PEI. For the content of Al, it decreased from 25.85% in the sol–gel film to 0.31% in the LBL film, which suggested that the LBL film was almost completely covered on the sol–gel film and the Al alloy was isolated from the outside environment. It was concluded that polyelectrolyte solutions including a certain concentration of PEI and PAA can be used to prepare polyelectrolyte multilayer composite films by chemical absorption, and it was expected to be an effective method for Al alloy protection. 

### 3.2. Chemical Composition of Different Films

XPS technology was utilized to analyze the chemical composition of different films on an Al alloy substrate. Figure 3a–c revealed the XPS survey spectra, as well as fitted curves for Al2p, O1s, C1s, or N1s for the anodic oxide film, sol–gel film, and LBL assembled film, respectively.

The XPS survey spectrum of the anodic oxide film sample (Figure 3a) mainly proved the presence of Al and O on the surface of film with the atomic fractions 21.19% and 44.81%, respectively. The characteristic peak of O1s was predominant in the survey scan, which indicated the composition of aluminum oxide. The peak binding energy of O1s was 530.80 eV, corresponding to Al_2_O_3_ in the formed anodice oxide film. The Al2p peak binding energy was corresponding to Al metal and Al oxide with the binding energies of 73.8 eV and 75.5 eV, individually.

The XPS survey spectrum of the sol–gel film sample mainly illustrated the presence of C, Al, and O on the surface of sol–gel film, as depicted in Figure 3b. The main elements on the surface of protective film were observed as O, Al, and C, and the atomic fraction was 51.64%, 24.12%, and 24.07%, respectively. There were a few changes in the binding energy of the O1s peak and Al2p peak of the sol–gel film, comparing with that of the anodic oxide film. The C1s peak binding energies were 284.60 eV, 283.50 eV, and 286.20 eV, corresponding to the C–H, C–O–Al, and C–O bonds. The O1s peak binding energies were 529.60 eV, 530.10 eV, 530.68 eV, and 531.71 eV, corresponding to oxide, O^2−^, Al_2_O_3_, and OH. The content of Al and O had a lower increment than that of the anodic oxide film. In addition to the element C, the amounts of the other impurity elements were relatively decreased, which illustrated that the sol–gel film was well combined with the anodic oxide film and played a certain role of filling and sealing.

In Figure 3c, the presence of C1s, N1s, and O1s was shown on the XPS survey spectrum of the LBL assembled film sample, with almost no Al element detected. The characteristic peaks of C1s, N1s, and O1s were rarely strong, and the atomic fractions were 65.90%, 13.28%, and 18.72%. The electron binding energy of oxygen was 530.8 eV, corresponding to the O–H bond in PAA. The electron binding energy of the N element was 397.90 eV and 399.50 eV, corresponding to the C–N and NH_2_–R bonds. The binding energy of C element was 283.50 eV, 284.0 eV, 284.83 eV, and 287.4 eV, corresponding to the (C*H_2_–CH_2_)_n_, C–H, C–O, and C–N bonds, which came from PEI and PAA, respectively. It was demonstrated that the LBL self-assembled film was successfully covered on the sol–gel film. No aluminum element was detected, indicating that the self-assembled film was entirely deposited onto the sol–gel film, and played a good sealing effect for the base material, which could prevent penetration of the corrosive ions and provide ideal protection for an Al alloy.

From the XPS results, it could be speculated that the Al atom could release three electrons and transform into Al^3+^, participating in the formation of the C–Al–O bond, and then improve the adhesion ability of the anodic oxide film. Furthermore, the presence of aluminum was not observed on the surface of the LBL film. Since the sol–gel film contains some Al^3+^ in it, the LBL film can be combined by a Vander Ed Ley interaction with the sol–gel film. After 10 cycles of self-assembly, the LBL film is formed by the adsorption of anion and cation, and few defects were formed on the surface of the film; so, no Al element was detected. The diagram of the protective mechanism for different films was illustrated in Figure 4.

### 3.3. Surface Roughness

The AFM scan in Figure 5 showed the topography of different films. Figure 5a was the AFM morphology of bare Al alloy surface, which included the surface height of the sample [39]. The higher the surface, the lighter the color on the graph. It could be observed that the surface of the Al alloy substrate was uneven and covered with some scratches. The difference between the highest and lowest location was 948.00 nm.

Figure 5b indicated the AFM of Al alloy immersed in the oxalic acid system for some time. The morphology revealed that the film surface was also uneven, but the polishing marks disappeared, indicating that a layer was grown on the Al surface accompanying with some lamellar and step-like structures. The peak–valley difference was 363.90 nm, which was smaller than that of the Al alloy matrix. For oxide film with a 1 × 1 μm^2^ scanning area, the average roughness (R_a_) and root mean square roughness (R_q_) were found to be 39.7 nm and 51.9 nm, individually. Therefore, the as-prepared anodic oxide film was successfully covered on the surface of the Al alloy substrate, and it would endow the Al alloy with a certain protective ability for preventing the corrosion medium.

Figure 5c showed the AFM morphology of the sol–gel film. The peak–valley difference was about 245.10 nm, which was smaller than that of the Al alloy substrate and the anodic oxide film. Scratches, cracks, as well as other visible defects were significantly reduced. At the scanning area of 1 × 1 μm^2^, R_a_ and R_q_ were 26.7 nm and 33.9 nm, respectively. The sol–gel film had changed the surface state of the anodic oxide film with a lot of small yurts replacing the lamellar and stepped structures. It maybe because some small molecules in the sol–gel materials could penetrate into the based film and fill the porous and microcracks; then, the surface roughness would be reduced and the corrosion resistance of the substrate was improved.

Figure 5d showed the AFM map of the LBL film. From the morphology, it could be seen that the peak–valley difference was about 152.00 nm, and R_a_ and R_q_ were 18.2 nm and 23.4 nm, respectively at the scanning area of 1 × 1 μm^2^. The film was more smooth and flat than that of the other films, and almost no pore structures could be observed by naked eyes. As the water-soluble polyelectrolyte was deposited alternatively from the anions to cations, positive and negative charges were adsorbed on the surface of the film by electrostatic binding force. The formed self-assembled film was dense, uniform, and smooth. The surface roughness was the smallest, which could create an ideal protection for the Al alloy matrix.

### 3.4. Anticorrosion Properties

The anticorrosion performance is a key factor in evaluating the possibilities of the protective film in fundamental research and practical applications [31]. The electrochemical behaviors of four different samples, such as the blank Al alloy, Al alloy with anodic oxide film, sol–gel film, and LBL film were determined in 3.5 wt % NaCl solution individually, and the results are illustrated in Figure 6.

In Figure 6, it can be seen that the radii of the capacitive arc for four types of samples were different, indicating various protective abilities of the protective films. For the blank Al alloy sample, it was the smallest, followed by an anodic oxide film, sol–gel film, and LBL samples. Meanwhile, in Figure 6b, the total system impedance followed the law of the blank sample, anodic oxide film, sol–gel film, and LBL film. Especially for the LBL film, the system impedance was higher than that for 10^8^ Ω·cm^2^, indicating an excellent protective performance compared with the other films. For the phase angle in Figure 6c, except for the blank Al alloy sample, all the other samples presented multi time-constant features. Figure 7 showed the diagram of equivalent circuit models used for fitting the measured EIS data, and the fitted results were illustrated in Table 1. For the blank Al alloy, the Nyquist spectrum presented only one capacitive semicircle. The surface of the blank Al alloy was abraded by the SiC sand paper, and no film was formed on the metal surface; therefore, the characterization of the determined curve was the charge-transfer process at the solution/alloy interface. Thus, the corrosion parameters should be fitted by Model 1, and the code could be described as R_s_(Q_dl_R_ct_). R_s_ denoted the solution resistance, Q_dl_ and R_ct_ represented the double-layer capacitance and the charge-transfer resistance of the interfacial double electric layer, respectively. Especially, in this work, the entire constant phase elements Q were used to model the capacitance because of the heterogeneity of the electrode surface. The impedance of Q is expressed as
(1)$$Z_{Q}=\frac{1}{Y_{0}{\left(j\omega \right)}^{n}}$$

Herein, Y_0_ and n are the coefficient and the exponent, respectively, ω is the angular frequency in rad s^−1^ (ω=2πf), and j is the imaginary unit with j^2^ = −1 [40,41]. For the anodic oxide film, the curve in Figure 6a1 showed a depressed capacitive semicircle. Essentially, the semicircle was formed by two overlapping semicircles, which were located respectively at high frequency and low frequency and were related to two electrochemical processes. The high-frequency semicircle was associated with the anodic oxide film, whereas the low-frequency semicircle was corresponded to the charge-transfer process. Since there were numerous defects in the anodic oxide film as shown in Figure 2b, the corrosive medium could penetrate the anodic oxide film directly through these cracks into the film/metal interface; then, it could result in various series connections of these two electrochemical processes, as displayed by Model 2 in Figure 7. In addition, the total model code could be expressed as the series circuit R_s_(Q_c_R_c_)(Q_dl_R_ct_), of which Q_c_ and R_c_ were the capacitance and resistance of the protective film, respectively.

For the sol–gel film, although the system impedance was higher than that of the anodic oxide film, the layer could not play a good protection role because of the small amount of micropores. Thus, Model 2 was also employed to imitate sol–gel film with two electrode processes.

For the LBL film, the system impedance reached 1.32 × 10^8^ Ω·cm^2^, which was significantly higher than that of the other films, suggesting that the corrosive ions could be effectively inhibited from participating in the electrochemical reaction [42]; that is, the LBL film could provide great protective effect for base metals. From Figure 6c, three time constants can be seen presenting in Bode plots for LBL film. So, the protective LBL layer should be regarded as a capacitor in the analyzing process of EIS plots, and the model code could be denoted as the parallel circuit R_s_(Q_LBL_(R_LBL_(Q_c_R_c_)(Q_dl_R_ct_))).

In summary, the total impedance of the three models could be calculated by the following equations:(2)$$\mathrm{Model}\;1:{\;Z}_{1}=R_{s}+\frac{1}{\frac{1}{R_{\mathrm{ct}}}+Y_{0\mathrm{dl}}{\left(j\omega \right)}^{n_{1}}}$$
(3)$$\mathrm{Model}\;2:{\;Z}_{2}=R_{s}+\frac{1}{\frac{1}{R_{c}}+Y_{0c}{\left(j\omega \right)}^{n^{2}}}+\frac{1}{\frac{1}{R_{\mathrm{ct}}}+Y_{0\mathrm{dl}}{\left(j\omega \right)}^{n1}}$$
(4)$$\mathrm{Model}\;3:\;Z_{3}=R_{s}+\frac{1}{Y_{0\mathrm{LBL}}{\left(j\omega \right)}^{n_{3}}+\frac{1}{R_{\mathrm{LBL}}+\frac{1}{\frac{1}{R_{c}}+Y_{0c}{\left(j\omega \right)}^{n^{2}}}+\frac{1}{\frac{1}{R_{\mathrm{ct}}}+Y_{0\mathrm{dl}}{\left(j\omega \right)}^{n1}}}}$$

The fitted results are illustrated in Table 1. Once the surface of the Al alloy was covered with different films, the electrochemical parameters changed accordingly. Obviously, the changing law of the R_c_ value followed the order of LBL > sol–gel > oxide film > blank Al alloy. The lowest R_c_ value of the blank Al alloy indicated that the erosion process occurred more easily compared to the other samples, indicating that corrosion was prone to be induced in this material. The R_c_ value increased as LBL films were deposited on the sol–gel film, confirming that the polyelectrolytes could remedy the sol–gel film defect and enhance its protection ability. A corrosive medium could be separated effectively by the LBL self-assembling film, and cannot penetrate into the Al alloy through the LBL film to cause corrosion. Generally, the inhibition efficiency (η) can be obtained from R_ct_ by the following formula [43]:(5)$$\eta =\left(1-\frac{R_{\mathrm{ct}}^{0}}{R_{\mathrm{ct}}}\right)\times 100\%$$
where R^0^_ct_ signifies the charge transfer resistance of the pure Al alloy, and R_ct_ represents the charge transfer resistance of the Al alloy with different protection film.

According to the results of η illustrated in Table 1, the united coating containing a sol–gel layer and LBL film had the highest inhibition efficiency, which was 98.1%, indicating a remarkable corrosion resistance performance. These results further proved that the LBL film could combine perfectly with the sol–gel film and then obtain a protective film with excellent resistance to external erosion.

### 3.5. The Polarization Curves Test

Electrochemical measurements were used to estimate the electrochemical parameters associated with the corrosion process occurring on the substrate covered with different films during the immersion time [44]. It was generally recognized that a lower corrosion current density and higher corrosion potential corresponded to a better corrosion resistance. Figure 8 depicted the potentiodynamic polarization curves for different samples. It could be found that the corrosion current density was changed after being treated by the protective technologies. The Al alloy with an anodic oxide film and sol–gel film had much better corrosion resistance than the blank samples. For the LBL sample, the current density was only 0.00316 μA/cm^2^, which was two orders of magnitude lower than that of the other samples, and the open circuit potential was −0.31 V, which was 200 mV higher than that of the sol–gel sample and 400 mV higher than that of the other two samples. Therefore, the LBL film was beneficial to repel and isolate the corrosion medium from attaching to the substrate due to the pores and discharge channels being blocked by it [45], which was in accordance with the results of EIS measurements.

### 3.6. Salt Spray Tests

The traditional neutral salt spray tests were used to estimate the protection performance for different films. The surface morphology of different samples after the salt spray test for 168 h are depicted in Figure 9. There were many cracks that appeared obviously on the surface of the blank Al alloy, as shown in Figure 9a. The surface of the blank Al alloy sample was covered with large and loose products, looking similar to an irregular mass or cell shape. The products increased in the thickness and spread outward to form a large area of erosion with time. Four elements, such as Al, Cl, O, and Na, were detected in the corrosion products, indicating that the aluminum oxide film was too fragile to withstand the invasion of chloride ions. Pitting corrosion was too easy to occur for this kind of materials.

For the Al alloy covering with anodic oxide film, the surface was also not smooth, and some microcracks could be observed (Figure 9b) after the salt spray test. However, the number of pitting pits was less than that of the blank Al alloy, and the cracks were tiny and distributed evenly. It indicated that when the anodic oxide film was put in the environment including chloride ions, the corrosion process would initiate from the weak grain boundary gradually and outwardly. According to the results of EDS analysis, the white corrosion product was aluminum chloride or carbide, which might be because the anodic oxide film was a typical porous structure. The Al alloy matrix was invaded rapidly by chloride ions through the pore structure to cause corrosion. With the accumulation of corrosion products and prolonging of corrosion time, the protective effect of the anodic oxide film was reduced. The porous structure of the anodic oxide film had a high adsorption capacity to trigger corrosion by absorbing moisture, salt, and corrosive media in the surrounding environment. Therefore, the anodic oxide film should be sealed to improve the corrosion resistance.

The surface morphology of the sol–gel films after the salt spray test for 168 h are depicted in Figure 9c. Some microcracks accompanied with nubby bumps appeared on the surface of the samples. However, the width of the cracks was significantly lower than those of the blank Al alloy and the anodic oxide film after the salt spray test. According to the results of EDS analysis, the content of the Cl element on the surface was only 2.97%, indicating that the salt spray corrosion was partly prevented by the sol–gel film and did not penetrate into the matrix. Although the sol–gel film was uniform and dense, there were still fine pores in macroscopic morphology. Since the radius of chloride ion was much less than the breadth of the cracks, they would easily penetrate into the substrate. Therefore, these defects of the sol–gel film under salt spray conditions made it unable to withstand the salt fog erosion over a long term, so the sol–gel film also needed further treatment by other technologies.

From Figure 9d, it can be seen that the corrosion degree of the LBL film was light, comparing with the other samples. The surface of the LBL film had high uniformity and smoothness from the macroscopic view. Less corrosion products and defects were found on the surface. The main elements for the LBL film after the salt spray test for 168 h were C and O. The content of Cl and Al was only 0.58% and 1.47%, respectively. Almost no chloride ions adhered on the surface, and the substrate of aluminum was not corroded during the process of the salt-spray test. Therefore, it could be concluded that the as-prepared LBL self-assembled film covered the surface of the sol–gel film ideally, blocking the eroding path of the corrosive medium and improving the corrosion resistance of the Al alloy.

## 4. Conclusions

In summary, a multilayered coating was successfully fabricated on a 2A12 aluminum alloy surface combining anodization, sol–gel, and LBL strategies. The anodic oxide film and sol–gel film can improve the anticorrosive abilities of the Al alloy, with the charge transfer resistant increased by one or two orders of magnitude compared to the blank Al alloy. However, the cracks were still observed in these two films after analysis by SEM, which would affect the long-term protective ability for the Al alloy. The LBL layer, which was proven to have a good sealing performance, could fill the pores and cracks effectively and improve the anticorrosion ability of the Al alloy with the coating resistance as high as 9.03 × 10^6^ Ω·cm^2^, and the system impedance is higher than 10^8^ Ω·cm^2^. After the salt spray test, the film, which was prepared by the LBL method, exhibited the best protective ability with less defects and a good surface state. It was believed that this kind of multilayer would have broad applications and bright prospects in marine corrosion and protection.

## Figures and Tables

**Figure 1 materials-13-00111-f001:**
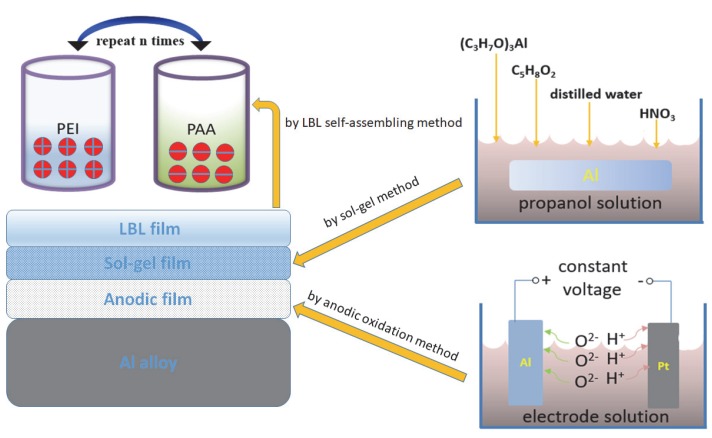
Schematic of the construction for protection film on Al alloy.

**Figure 2 materials-13-00111-f002:**
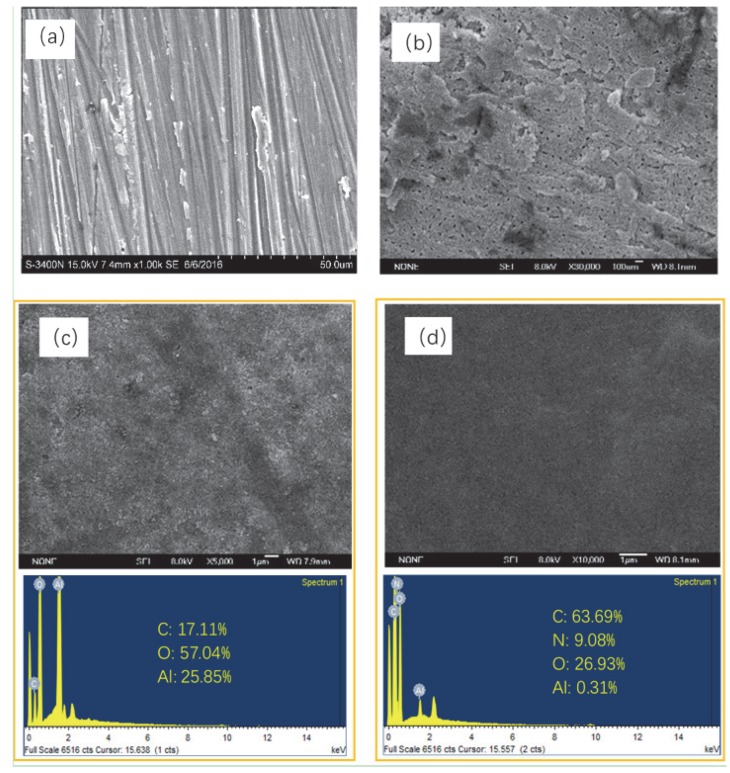
SEM images and energy dispersive X-ray spectrum (EDS) results of (**a**) bare Al alloy, (**b**) anodic oxide film, (**c**) sol–gel film, and (**d**) layer-by-layer (LBL) film.

**Figure 3 materials-13-00111-f003:**
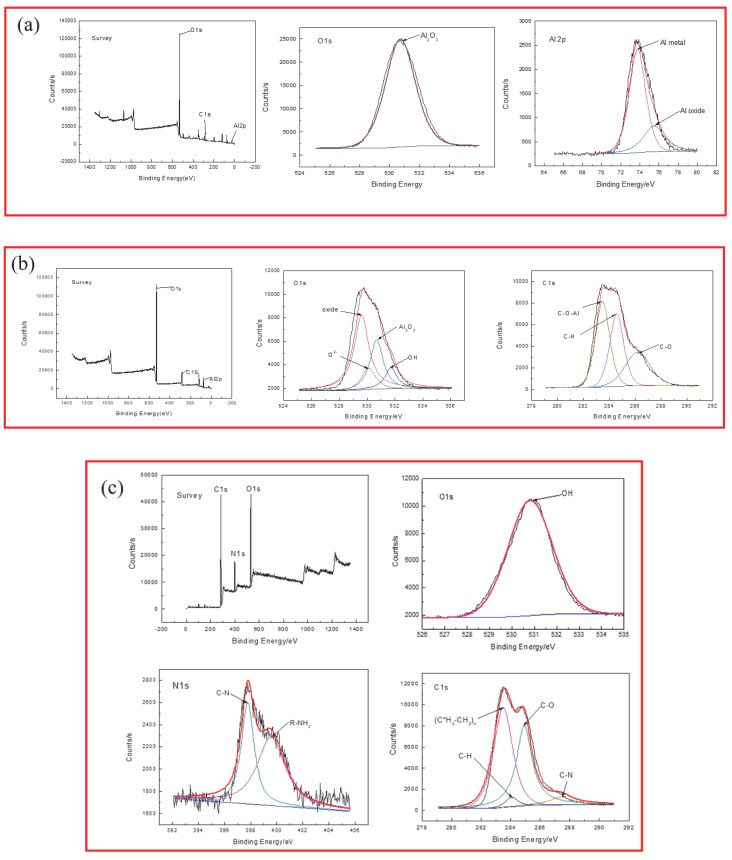
X-ray photoelectron spectroscopy spectrum of different films: (**a**) anodic oxide film, (**b**) sol–gel film, and (**c**) LBL film.

**Figure 4 materials-13-00111-f004:**
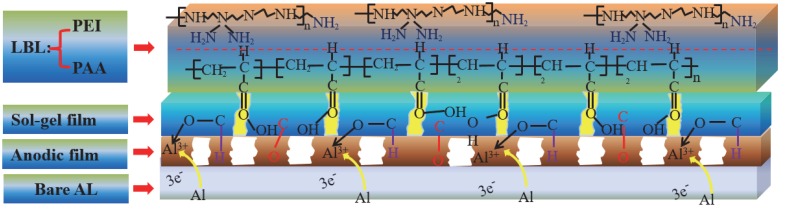
Schematic diagram of protection mechanism.

**Figure 5 materials-13-00111-f005:**
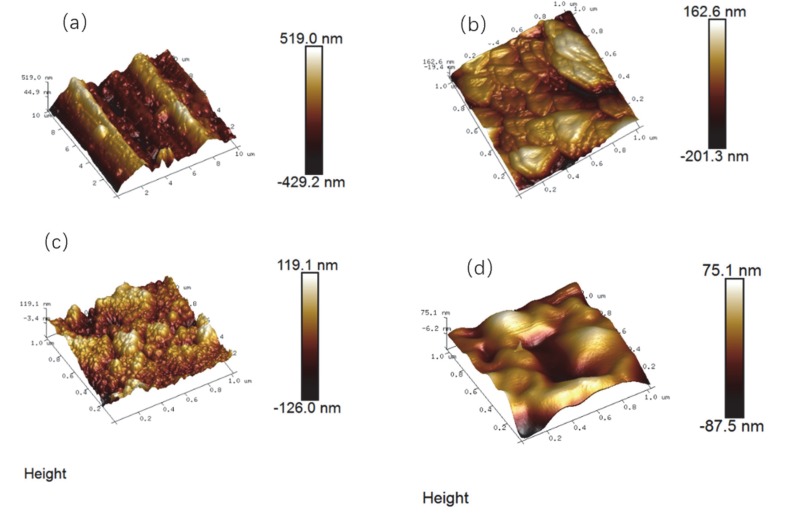
Atomic force microscope (AFM) tapping mode 3D images of (**a**) the Al alloy, (**b**) anodic oxide film, (**c**) sol–gel film, and (**d**) LBL film.

**Figure 6 materials-13-00111-f006:**
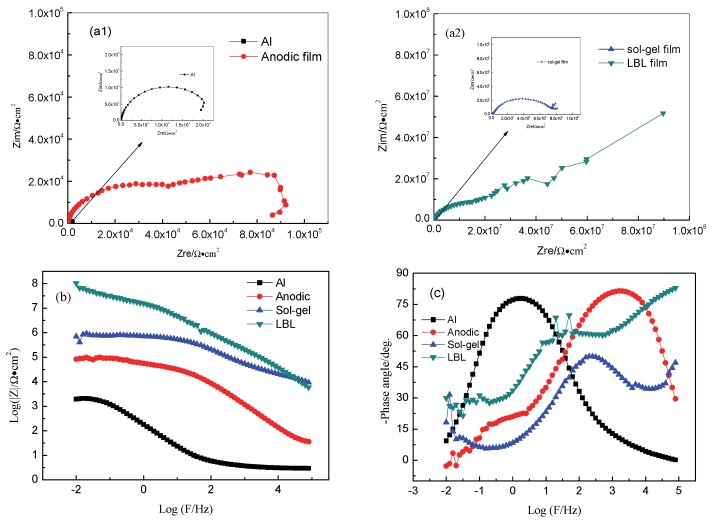
Electrochemical impedance spectroscopy plots and the fitting curves of Al alloy samples with different film. (**a1**) and (**a2**) Nyquist plots; (**b**) and (**c**) Bode plots.

**Figure 7 materials-13-00111-f007:**
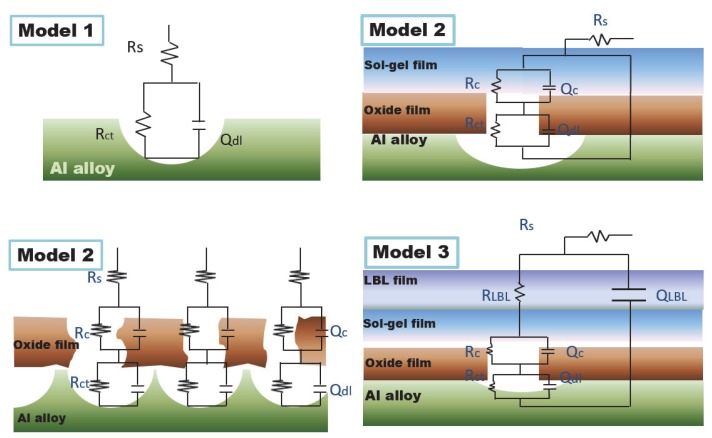
Equivalent models of EIS plots for Al alloy samples with different films.

**Figure 8 materials-13-00111-f008:**
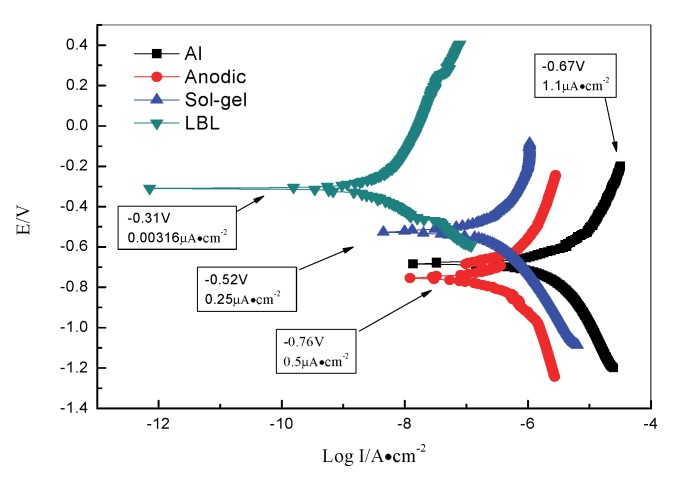
Polarization curves of Al alloy, anodic oxide film, sol–gel film, and LBL film.

**Figure 9 materials-13-00111-f009:**
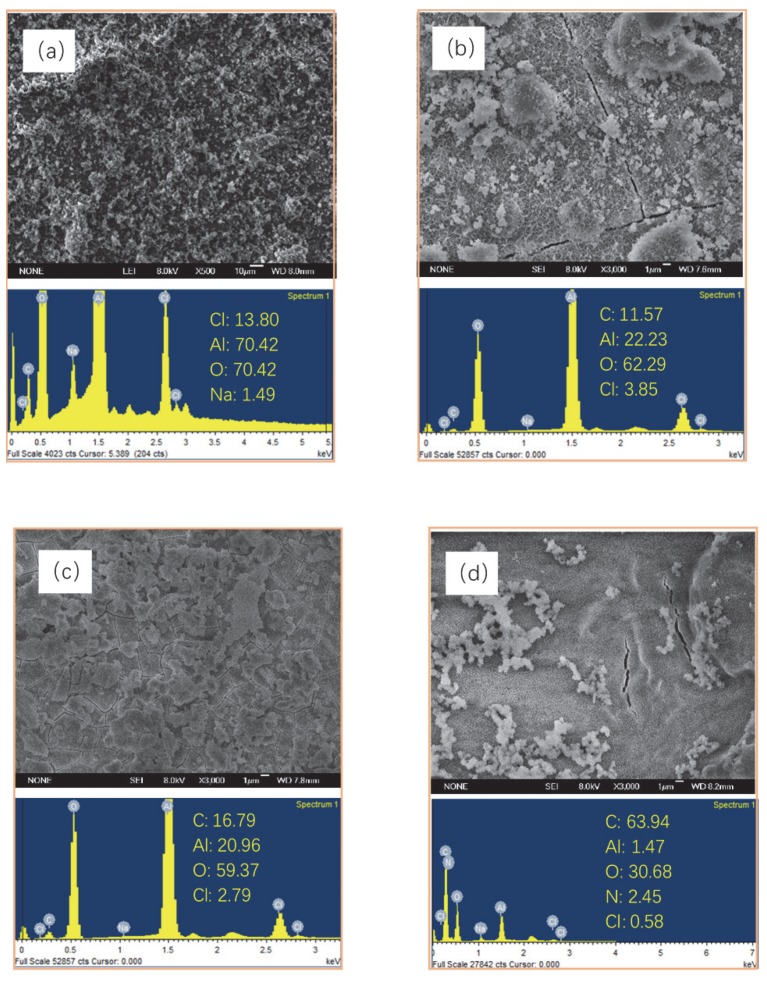
The surface morphology of (**a**) the blank Al alloy, (**b**) anodic oxide film, (**c**) sol–gel film, (**d**) LBL film, and EDS images after the salt spray test for 168 h.

**Table 1 materials-13-00111-t001:** Fitting results of EIS for Al alloy with different films.

Sample	Q_dl_	n	R_ct_	Q_c_	n	R_c_	Q_LBL_	n	R_LBL_	η
	Y_0dl_ Ω^−1^cm^−2^s^n^		Ω·cm^2^	Y_0dl_ Ω^−1^cm^−2^s^n^		Ω·cm^2^	Y_0LBL_ Ω^−1^cm^−2^s^n^		Ω·cm^2^	%
Al alloy	0.00106	0.88	2446	\	\	\	\	\	\	
Oxide film	2.95 × 10^−7^	0.93	3.25 × 10^4^	5.94 × 10^−6^	0.83	6.29 × 10^4^	\	\	\	92.5
Sol–gel film	4.42 × 10^−9^	0.75	3.58 × 10^4^	4.74 × 10^−8^	0.69	7.54 × 10^5^	\	\	\	93.2
LBL film	0.76 × 10^−11^	0.93	1.30 × 10^5^	0.79 × 10^−10^	0.87	9.03 × 10^6^	0.45 × 10^−9^	0.52	1.32 × 10^8^	98.1

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
