# Peer review of "Perfect Combination of LBL with Sol–Gel Film to Enhance the Anticorrosion Performance on Al Alloy under Simulated and Accelerated Corrosive Environment"

_materials, 2019, doi:10.3390/ma13010111_

Round 1
Reviewer 1 Report
The paper is well written. The results are presented in a clear way and discussion and significance of the work are properly evidenced. In my opinion, the paper is suitable for publication essentially in the present form, although some minor corrections should be made. In particular, I would suggest the authors to spend some words to explain the exact reasons behind the choice of PEI and PAA as polyelectrolytes to build the multi layer coating system. Is this choice based on previous anticorrosion literature or is it an original approach developed by the authors for Al alloys?
Other very minor remarks follow:
Ref 11: spelling error : "cost" to replace "coat".
Refs 19 and 22: they have the same spelling error: organic-inorganic to replace organiceinorganic.
Ref. 26: this reference does not seem appropriate with the text; please give a check.
pag 11: again a spelling error : -0.31V to replace 0.31V.
Author Response
Reviewer 1
Thank you very much for all of your valuable suggestions in improving the quality of this paper. Thank you!
Question 1.The exact reasons behind the choice of PEI and PAA as polyelectrolytes to build the multi layer coating system.
Response: Some reasons had been added in line 55-57, and the corresponding literature had been added as [33].
Question 2. Ref 11: spelling error : "cost" to replace "coat".
Response: Revised.
Question 3. Refs 19 and 22: they have the same spelling error: organic-inorganic to replace organiceinorganic.
Response: Revised.
Question 4. Ref. 26: this reference does not seem appropriate with the text; please give a check.
Response: This reference is a book, maybe it is not very appropriate with the text, so now, I replaced it with another one, as shown below.
26.Song X, Meng F, Kong M, Liu Z, Huang L, Zheng X, Zeng Y(2017), Relationship between cracks and microstructures in APS YSZ coatings at elevated temperatures, Mater Charact,131:277-284.
Question 5. Pag 11: again a spelling error : -0.31V to replace 0.31V.
Response:Revised.
Reviewer 2 Report
The present manuscript entitled “Perfect combination of LBL with sol-gel film to enhance the anticorrosion performance on Al alloy under simulated and accelerated corrosive environment” falls under the scope area of Materials journal. However, it needs revision for publication in journal. My following comments are required to justify and modify the manuscript accordingly:
Lines 41-46 The idea that the aluminum oxide offers a poor protection against corrosion is present in the text, but the writing is a bit confusing, I suggest that the authors revise the English. This suggestion should be taking into consideration for the entire paper with particular focus on introduction and discussion sections. Line 48 “marine environment” - not only in the marine environment, Al alloys are used in several fields (as the authors well mention in the beginning of introduction), if the present work is focused in this field in particular, this should be mentioned. Figure 1 - I suggest to improve the description of the figure. As it is, is not straightforward to understand and extract all the information that the figure has. Line 217 “The XPS survey spectra of LBL assembled film” , the mentioned spectra is not presented. Is in supporting information? If so please write. If the authors choose not to presented it, it should be informed. In surface roughness section the authors should include the RMS values for direct comparison of roughness parameter. Line 378-379 “But the width and depth of cracks were significantly lower”, How can you tell just by looking at the images? The width I understand but you cannot infer the depth from there. Lines 402-403 “However, the cracks were still observed in these two films after analyzed by SEM and EIS”, The cracks can be seen by SEM not by EIS, the EIS results can support the observations of the SEM images.
Author Response
Reviewer 2
Thank you very much for valuable suggestions to improve of our manuscript. We had considered deeply and revised the questions carefully.
Question 1. Lines 41-46 The idea that the aluminum oxide offers a poor protection against corrosion is present in the text, but the writing is a bit confusing, I suggest that the authors revise the English. This suggestion should be taking into consideration for the entire paper with particular focus on introduction and discussion sections.
Response: Revised.
Question 2. Line 48 “marine environment” - not only in the marine environment, Al alloys are used in several fields (as the authors well mention in the beginning of introduction), if the present work is focused in this field in particular, this should be mentioned.
Response: Revised.
Question 3. Figure 1 - I suggest to improve the description of the figure. As it is, is not straightforward to understand and extract all the information that the figure has.
Response: Revised.
Question 4. Line 217 “The XPS survey spectra of LBL assembled film” , the mentioned spectra is not presented. Is in supporting information? If so please write. If the authors choose not to presented it, it should be informed. In surface roughness section the authors should include the RMS values for direct comparison of roughness parameter.
Response: I have added the Survey image in Fig.3c. And the values of the roughness parameter, such as Ra and Rq, were also added in text for comparison.
Question 5. Line 378-379 “But the width and depth of cracks were significantly lower”, How can you tell just by looking at the images? The width I understand but you cannot infer the depth from there.
Response: Revised.
Question 6. Lines 402-403 “However, the cracks were still observed in these two films after analyzed by SEM and EIS”, The cracks can be seen by SEM not by EIS, the EIS results can support the observations of the SEM images.
Response: Revised.